# AMRL: Aggregated Memory For Reinforcement Learning

**Jacob Beck**[*]**, Kamil Ciosek, Sam Devlin, Sebastian Tschiatschek, Cheng Zhang, Katja Hofmann**
Microsoft Research, Cambridge, UK
[*]`Jacob_Beck@alumni.brown.edu, Firstname.Lastname@microsoft.com`

## Abstract

In many partially observable scenarios, Reinforcement Learning (RL) agents must rely on long-term memory in order to learn an optimal policy. We demonstrate that using techniques from natural language processing and supervised learning fails at RL tasks due to stochasticity from the environment and from exploration. Utilizing our insights on the limitations of traditional memory methods in RL, we propose *AMRL*, a class of models that can learn better policies with greater sample efficiency and are resilient to noisy inputs. Specifically, our models use a standard memory module to summarize short-term context, and then aggregate all prior states from the standard model without respect to order. We show that this provides advantages both in terms of gradient decay and signal-to-noise ratio over time. Evaluating in Minecraft and maze environments that test long-term memory, we find that our model improves average return by 19% over a baseline that has the same number of parameters and by 9% over a stronger baseline that has far more parameters.

## 1 Introduction

We address the problem of reinforcement learning (RL) in tasks that require long-term memory. While many successes of Deep RL were achieved in settings that are (near) fully observable, such as Atari games (Mnih et al., 2015), partial observability requires memory to recall prior observations that indicate the current state. Relying on full observability severely limits the applicability of such approaches. For example, many tasks in virtual and physical environments are naturally observed from a first-person perspective (Oh et al., 2016), which means that an agent may need to seek out and remember task-relevant information that is not immediately observable without directly observing the entire environment. Recent research has started to address this issue, but effective learning in RL settings with long sequential dependencies remains a key challenge in Deep RL (Oh et al., 2016; Stepleton et al., 2018; Parisotto & Salakhutdinov, 2018).

The currently most common approach to RL in partially observable settings relies on models that use memory components that were originally developed for tasks like those that occur in natural language processing (NLP), e.g., LSTMs (Hochreiter & Schmidhuber, 1997) and GRUs (Cho et al., 2014). Hausknecht & Stone (2015) first demonstrated benefits of LSTMs in RL tasks designed to test memory, and these and similar approaches have become common in Deep RL (Wang et al., 2016), including multi-agent RL (Rashid et al., 2018; Foerster et al., 2017).

In this work, we demonstrate that the characteristics of RL can severely impede learning in memory models that are not specifically designed for RL, and propose new models designed to tackle these challenges. For example, LSTMs excel in NLP tasks where the order of observations (characters or words) is crucial, and where influence between observations decays quickly with distance. Contrast this with a hypothetical RL example where an agent must discover a hidden passcode to escape a locked dungeon. The order of observations is highly dependent on the agent's path through the dungeon, yet when it reaches the door, only its ability to recall the passcode is relevant to escaping the dungeon, irrespective of when the agent observed it and how many observations it has seen since.

Figure 1 illustrates the problem. When stochasticity is introduced to a memory task, even simply as observation noise, the sample efficiency of LSTMs decreases drastically. We show that this problem occurs not just for LSTMs, but also for stacked LSTMs and DNCs (Graves et al., 2016; Wayne et al., 2018), which have been widely applied in RL, and we propose solutions that address this problem.

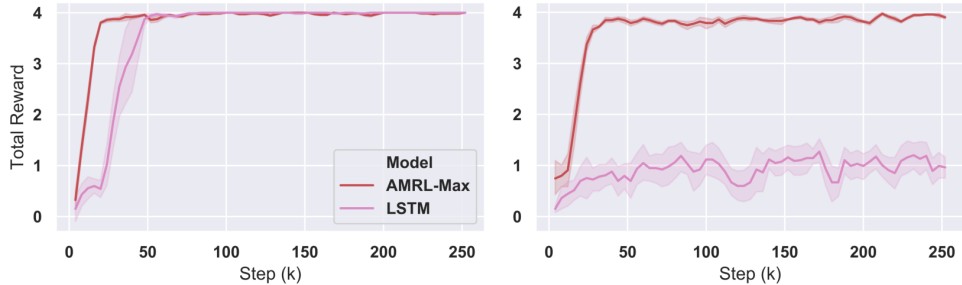

Figure 1: Our AMRL-Max model compared to a standard LSTM memory module (Hochreiter & Schmidhuber, 1997) trained on a noise-free memory task (**T-L**, left) and the same task with observational noise (**T-LN**, right). In both cases, the agent must recall a signal from memory after navigating through a corridor. LSTM completely fails with the introduction of noise, while AMRL-Max learns rapidly. (68% confidence interval over 5 runs, as for all plots.)

We make the following three contributions. First, in Section 3, we introduce our approach, *AMRL*. AMRL augments memory models like LSTMs with *aggregators* that are substantially more robust to noise than previous approaches. Our models combine several innovations which jointly allow the model to ignore noise while maintaining order-variant information as needed. Further, AMRL models maintain informative gradients over very long horizons, which is crucial for sample-efficient learning in long-term memory tasks (Pascanu et al., 2012; Bakker, 2001; Wierstra et al., 2009).

Second, in Section 4, we systematically evaluate how the sources of noise that affect RL agents affect the sample efficiency of AMRL and baseline approaches. We devise a series of experiments in two domains, (1) a symbolic maze domain and (2) 3D mazes in the game Minecraft. Our results show that AMRL can solve long-term memory tasks significantly faster than existing methods. Across tasks our best model achieves an increase in final average return of 9% over baselines with far more parameters and 19% over LSTMs with the same number of parameters.

Third, in Section 6 we analytically and empirically analyze the characteristics of our proposed and baseline models with the aim to identify factors that affect performance. We empirically confirm that AMRL models are substantially less susceptible to vanishing gradients than previous models. We propose to additionally analyze memory models in terms of the signal-to-noise ratio achieved at increasing distances from a given signal, and show that AMRL models can maintain signals over many timesteps. Jointly, the results of our detailed analysis validate our modeling choices and show why AMRL models are able to effectively solve long-term memory tasks.

## 2 RELATED WORK

**External Memory in RL.** In the RL setting, work on external memory models is most relevant to our own. Oh et al. (2016) introduce a memory network to store a fixed number of prior memories after encoding them in a latent space and validate their approach in Minecraft, however the models are limited to fixed-length memories (i.e., the past 30 frames). The Neural Map of Parisotto & Salakhutdinov (2018) is similar to our work in that it provides a method in which past events are not significantly harder to learn than recent events. However, it is special-cased specifically for agents on a 2D grid, which is more restrictive than our scope of assumptions. Finally, the Neural Turing Machine (NTM) (Graves et al., 2014) and its successor the Differentiable Neural Computer (DNC) (Graves et al., 2016) have been applied in RL settings. They use an LSTM controller and attention mechanisms to explicitly write chosen memories into external memory. Unlike the DNC which is designed for algorithmic tasks, intentionally stores the order of writes, and induces sparsity in memory to avoid collisions, we write memories into order-invariant aggregation functions that provide benefits in noisy environments. We select the DNC, the most recent and competitive prior approach, for baseline comparisons.

**Other Memory in RL.** A second and orthogonal approach to memory in Deep RL is to learn a separate policy network to act as a memory unit and decide which observations to keep. These approaches are generally trained via policy gradient instead of back-propagation through time (BPTT) (Peshkin et al., 1999; Zaremba & Sutskever, 2015; Young et al., 2018; Zhang et al., 2015; Han et al.,

2019). These approaches are often difficult to train and are orthogonal to our work which uses BPTT. The Low-Pass RNN (Stepleton et al., 2018) uses a running average similar to our models. However, they only propagate gradients through short BPTT truncation window lengths. In fact they show that LSTMs outperform their method when the window size is the whole episode. Since we are propagating gradients through the whole episode, we use LSTMs as a baseline instead. Li et al. (2015); Mirowski et al. (2017); Wayne et al. (2018) propose the use of self-supervised auxiliary losses, often relating to prediction of state transitions, to force historical data to be recorded in memory. Along this line, model-based RL has also made use of memory modules to learn useful transition dynamics instead of learning a policy gradient or value function (Ke et al., 2019; Ha & Schmidhuber, 2018). These are orthogonal and could be used in conjunction with our approach, which focuses on model architecture. Finally, several previous works focus on how to deal with storing initial states for truncated trajectories in a replay buffer (Kapturowski et al., 2019; Hausknecht & Stone, 2015).

**Memory in Supervised Learning.** In the supervised setting, there has also been significant work in memory beyond LSTM and GRUs. Similar to our work, Mikolov et al. (2014), Oliva et al. (2017), and Ostmeyer & Cowell (2019) use a running average over inputs of some variety. Mikolov et al. (2014) use an RNN in conjunction with running averages. However, Mikolov et al. (2014) use the average to provide context to the RNN (we do the inverse), and all use an exponential decay instead of a non-decaying average. Additionally, there have been myriad approaches attempting to extend the range of RNNs, that are orthogonal to our work, given that any could be used in conjunction with our method as a drop-in replacement for the LSTM component (Le et al., 2015; Arjovsky et al., 2016; Krueger & Memisevic, 2016; Belletti et al., 2018; Trinh et al., 2018). Other approaches for de-noising and attention are proposed in (Kolbaek et al., 2017; Wöllmer et al., 2013; Vaswani et al., 2017; Lee et al., 2018) but have runtime requirements that would be prohibitive in RL settings with long horizons. Here, we limit ourselves to methods with $O(1)$ runtime per step.

## 3 METHODS

### 3.1 PROBLEM SETTING

We consider a learning agent situated in a partially observable environment denoted as a Partially Observable Markov Decision Process (POMDP) (Kaelbling et al., 1998). We specify this process as a tuple of $(\mathcal{S}, \mathcal{A}, \mathcal{R}, \mathcal{P}, \mathcal{O}, \Omega, \gamma)$. At time-step $t$, the agent inhabits some state, $s_t \in \mathcal{S}$, not observable by the agent, and receives some observation as a function of the state $o_t \in \Omega \sim \mathcal{O}(o_t|s_t)\colon \Omega \times \mathcal{S} \to \mathbb{R}_{\geq 0}$. $\mathcal{O}$ is known as the *observation function* and is one source of stochasticity. The agent takes some action $a_t \in \mathcal{A}$. The POMDP then transitions to state $s_{t+1} \sim \mathcal{P}(s_{t+1}|s_t, a_t)\colon \mathcal{S} \times \mathcal{A} \times \mathcal{S} \to \mathbb{R}_{\geq 0}$, and the agent receives reward $r_t = \mathcal{R}(s_t, a_t)\colon \mathcal{S} \times \mathcal{A} \to \mathbb{R}$ and receives a next observation $o_{t+1} \sim \mathcal{O}$ upon entering $s_{t+1}$. The *transition function* $\mathcal{P}$ also introduces stochasticity. The sequence of prior observations forms an observation trajectory $\tau_t \in \Omega^t \equiv \mathcal{T}$. To maximize $\sum_{t=0}^{t=\infty} \gamma^t r_t$, the agent chooses each discrete $a_t$ from a stochastic, learned *policy* conditioned on trajectories $\pi(a_t|\tau_t)\colon \mathcal{T} \times \mathcal{A} \to [0, 1]$. Given $\mathcal{P}$, $\mathcal{O}$, and $\pi$ itself are stochastic, $\tau_t$ can be highly stochastic, which we show can prevent learning.

### 3.2 MODEL

In this section we introduce our model *AMRL*, and detail our design choices. At a high level, our model uses an existing memory module to summarize context, and extends it to achieve desirable properties: robustness to noise and informative gradients.

**LSTM base model.** We start from a standard memory model as shown in Figure 2a. We use the base model to produce a contextual encoding of the observation $o_t$ that depends on the sequence of prior observations. Here, we use several feed-forward layers, $FF_1$ (defined in A.5), followed by an LSTM layer (defined in A.6):

$$\boldsymbol{e}_t = FF_1(\boldsymbol{o}_t) \text{ // The encoded observation}$$
$$\boldsymbol{h}_t = LSTM_t(\boldsymbol{e}_t) \text{ // The output of the LSTM}$$

Previous work proposed a stacked approach that combines two (or more) LSTM layers (Figure 2b) with the goal of learning higher-level abstractions (Pascanu et al., 2013; Mirowski et al., 2017).

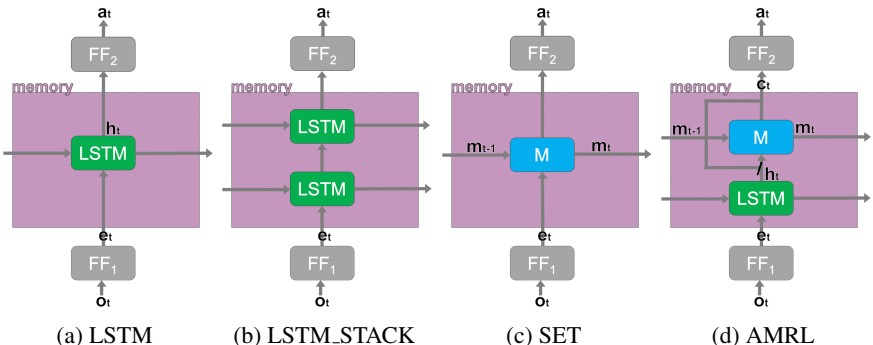

Figure 2: Model Architectures. AMRL (d) extends LSTMs (a) with SET based aggregators (c).

Table 1: Definition of our AVG, SUM, and MAX aggregators and their key properties (see text).

| Aggregator | Definition | Jacobian | ST Jacobian | SNR |
|---|---|---|---|---|
| AVG | $\mathbf{avg}_t = \frac{1}{t}\sum_{i=0}^{i=t} \boldsymbol{x}_i$ | $\frac{d\mathbf{avg}_t}{d\boldsymbol{x}_i} = \frac{1}{t}\boldsymbol{I}$ | $\boldsymbol{I}$ | $s_0^2/(tVar(n))$ |
| SUM | $\mathbf{sum}_t = \sum_{i=0}^{i=t} \boldsymbol{x}_i$ | $\frac{d\mathbf{sum}_t}{d\boldsymbol{x}_i} = \boldsymbol{I}$ | $\boldsymbol{I}$ | $s_0^2/(tVar(n))$ |
| MAX | $\mathbf{max}_t = \max_{i=0}^{i=t} \mathbf{x_i}$ | $\mathbb{E}[\frac{d\mathbf{max}_t}{d\boldsymbol{x}_i}] = \frac{1}{t}\boldsymbol{I}$ | $\boldsymbol{I}$ | $\geq \frac{s_0^2}{(|\Omega|-1)max(\Omega)^2}$ |

A key limitation of both LSTM and LSTM_STACK approaches is susceptibility to noise. Noise can be introduced in several ways, as laid out in Section 3.1. First, observation noise introduces variance in the input $o_t$. Second, as motivated by our introductory example of an agent exploring a dungeon, variance on the level of the trajectory $\tau_t$ is introduced by the transition function and the agent's behavior policy. Recall that in our dungeon example, the agent encounters many irrelevant observations between finding the crucial passcode and arriving at the door where the passcode allows escape. This variance in $\tau_t$ generally produces variance in the output of the function conditioning on $\tau_t$. Thus, although the first part of our model makes use of an LSTM to encode previous inputs, we expect the output, $\boldsymbol{h}_t$ to be sensitive to noise.

**Aggregators.** To address the issue of noise highlighted above, we introduce components designed to decrease noise by allowing it to cancel. We call these components *aggregators*, labeled $M$ in Figures 2c and 2d. An aggregator is a commutative function that combines all previous encodings $\boldsymbol{h}_t$ in a time-independent manner. Aggregators are computed dynamically from their previous value:

$$\boldsymbol{m}_t = g(\boldsymbol{m}_{t-1}, \boldsymbol{h}_t[:\frac{1}{2}]) \ \text{ // The aggregated memory}$$

where $\boldsymbol{h}_t[:\frac{1}{2}]$ denotes the first half of $\boldsymbol{h}_t$, and $g()$ denotes the aggregator function, the choices of which we detail below. All proposed aggregators can be computed in constant time, which results in an overall memory model that matches the computational complexity of LSTMs. This is crucial for RL tasks with long horizons.

In this work we consider the SUM, AVG, and MAX aggregators defined in Table 1. All three are easy to implement in standard deep learning frameworks. They also have desirable properties in terms of gradient flow (Jacobian in Table 1 and signal-to-noise ratio (SNR)) which we detail next.

**Aggregator signal-to-noise ratio (SNR).** Our primary design goal is to design aggregators that are robust to noise from the POMDP, i.e., variation due to observation noise, behavior policy or environment dynamics. For example, consider the outputs from previous timesteps, $\boldsymbol{h}_t$, to be i.i.d. vectors. If we use the average of all $\boldsymbol{h}_t$ as our aggregators, then the variance will decrease linearly with $t$. To formally and empirically assess the behavior of memory models in noisy settings, we propose the use of the signal-to-noise ratio or SNR (Johnson, 2006). The SNR is a standard tool for assessing how well a system maintains a given signal, expressed as the ratio between the signal (the information stored regarding the relevant information) and noise. In this section, to maintain flow, we simply state the SNR that we have analytically derived for the proposed aggregators in Table 1. We note that the SNR decays only linearly in time $t$ for SUM and AVG aggregators, which is empirically slower than the baselines, and has a bound independent of time for the MAX aggregator. We come back to this topic in Section 6, where we describe the derivation in detail, and provide further empirical results that allow comparison of all proposed and baseline methods.

**Aggregator gradients.** In addition to making our model robust to noise, our proposed aggregators can be used to tackle vanishing gradients. For example, the sum aggregator can be viewed as a residual skip connection across time. We find that several aggregators have this property: given that the aggregator does not depend on the order of inputs, the gradient does not decay into the past for a fixed-length trajectory. We can show that for a given input $x_i$ and a given output $o$, the gradient $\frac{do_t}{dx_i}$ (or expected gradient) of our proposed aggregators does not decay as $i$ moves away from $t$, for a given $t$. We manually derived the Jacobian column of Table 1 to show that the gradient does not depend on the index $i$ of a given past input. We see that the gradient decays only linearly in $t$, the current time-step, for the AVG and MAX aggregators. Given that the gradients do not vanish when used in conjunction with $h_t$ as input, they provide an immediate path back to each $h_t$ through which the gradient can flow.

**SET model.** Using an aggregator to aggregate all previous $o_t$ yields a novel memory model that we term *SET* (Figure 2c). This model has good properties in terms of SNR and gradient signal as shown above. However, it lacks the ability to maintain order-variant context. We address this limitation next, and include the SET model as an ablation baseline in our experiments.

**Combining LSTM and aggregator.** In our *AMRL* models, we combine our proposed aggregators with an LSTM model that maintains order-dependent memories, with the goal to obtain a model that learns order-dependent and order-independent information in a manner that is data efficient and robust to noise. In order to achieve this, we reserve certain neurons from $h_t$, as indicated by ′/′ in Figure 2d. We only apply our aggregators to one half of $h_t$. Given that our aggregators are commutative, they lose all information indicating the context for the current time-step. To remedy this, we concatenate the other half of $h_t$ onto the aggregated memory. The final action is then produced by several feed-forward layers, $FF_2$ (defined in A.5):

$$c_t = [h_t[\frac{1}{2} :]||f_t(m_t)] \text{ // The final context}^1$$

$$a_t = FF_2(c_t) \text{ // The action output by the network}$$

**Straight-through connections.** We showed above that our proposed aggregators provide advantages in terms of maintaining gradients. Here, we introduce a further modification that we term *straight-through* (ST) connections, designed to further improve gradient flow. Given that our proposed aggregators are non-parametric and fairly simple functions, we find that we can deliberately modify their Jacobian as follows. We pass the gradients *straight through* the model without any decay. Thus, in our ST models, we modify the Jacobian of $g$ and set it to be equal to the identity matrix (as is typical for non-differentiable functions). This prevents gradients from decaying at all, as seen in the ST Jacobian column of Table 1.

Our proposed models combine the components introduced above: **AMRL-Avg** combines the AVG aggregator with an LSTM, and uses a straight-through connection. Similarly, **AMRL-Max** uses the MAX aggregator instead. Below we also report ablation results for **SET** which uses the AVG aggregator without an LSTM or straight-through connection. In Section 5 we will see that all components of our model contribute to dramatically improved robustness to noise compared to previous models. We further validate our design choices by analyzing gradients and SNR of all models in Section 6.

## 4 EXPERIMENTS

We examine the characteristics of all proposed and baseline models through a series of carefully constructed experiments. In all experiments, an RL agent (we use PPO (Schulman et al., 2017), see Appendix A.5 for hyper-parameters) interacts with a maze-like environment. Similar mazes were proposed by Bakker (2001) and Wierstra et al. (2009) in order to evaluate long term memory in RL agents. Visual Minecraft mazes have been used to evaluate fixed-length memory in Oh et al. (2016).

We compare agents that use one of our proposed AMRL models, or one of the baseline models, as drop-in replacement for the combined policy and value network. (A single network computes both by modifying the size of the output.) Baselines are detailed in Section 4.3.

---

[1]Note that $f_t$ is applied to the vector $m_t$. Although the sum and max aggregator can be computed using the function $g(x, y) = sum(x, y)$ and $g(x, y) = max(x, y)$ respectively, the average must divide by the time-step. Thus, we set $f_t(x) = \frac{x}{t}$ for the average aggregator and $f_t(x) = x$ otherwise.

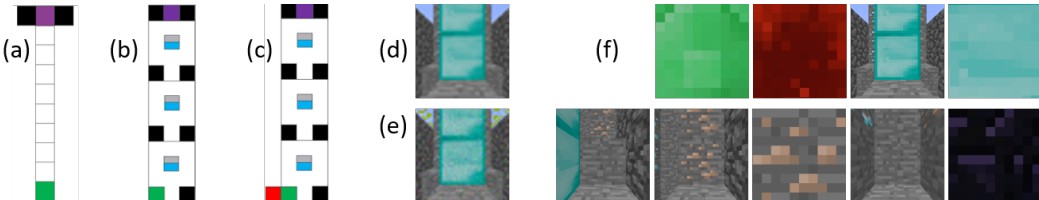

Figure 3: Overview of tasks used in our experiments: (a) Length-10 variant of TMaze (the full task has length 100); (b) a 3-room variant of **MC-LS** (Full Minecraft tasks have 10 or 16 rooms.); (c) **MC-LSO**; (d) original (**MC-LS(O)**) and (e) noisy (**MC-LSN**) Minecraft observation; (e) sample optimal trajectory (left-right, top-down) through a 1-room variant of **MC-LSO**.

## 4.1 TMAZE TASKS

In this set of tasks, an agent is placed in a T-shaped maze, at the beginning of a long corridor, as shown in Figure 3(a) (here green indicates both the start state and the indicator color). The agent must navigate to the end of the corridor (purple) where it faces a binary decision task. It must step left or right according to the start *indicator* it observed which requires memory to retain. At each time-step, the agent receives its observation as a vector, encoding whether the current state is at the start, in the corridor, or at the end. In the start state, the color of the indicator is also observed.

Our experiments use the following variants of this task (see Appendix A.1 for additional detail):

**TMaze Long (T-L)** Our base task reduces to a single decision task: the agent is deterministically stepped forward until it reaches the end of the corridor where it must make a decision based on the initial indicator. Corridor and episode length is 100. Reward is 4 for the correct action, and -3 otherwise. This task eliminates exploration and other noise as a confounding factor and allows us to establish base performance for all algorithms.

**TMaze Long Noise (T-LN)** To test robustness to noise, observations are augmented by a random variable $n \in \{-1, 1\}$, sampled uniformly at random. The variable $n$ is appended to the obervation, which is vector-valued. Other details remain as in **T-L**.

**TMaze Long-Short (T-LS)** Our hardest TMaze task evaluates whether additional short-term tasks interfere with a memory model trying to solve a long-term memory task. We add an intermediate task: we append $n \in \{-1, 1\}$, sampled uniformly at random, to the input and only allow the agent to progress forward if its discrete action $a \in \{-1, 1\}$ matches. Corridor length is still 100. A maximum episode length of 150 is imposed given the introduction of exploration.

## 4.2 MINECRAFT MAZE TASKS

In order to test that our approach generalizes to high dimensional input, we create multiple Minecraft environments using the open-source Project Malmo (Johnson et al., 2016). Compared to the previous setup, we replace fully-connected $FF_1$ layers with convolutional layers to allow efficient feature learning in the visual space (see Appendix A.5 for details). As in Oh et al. (2016), we use discrete actions. We allow movement in: {North, East, South West}. We use the following task variants:

**MC Long-Short (MC-LS)** Our first Minecraft environment tests agents' ability to learn short and long-term tasks - which adds the need to process video observations to the **T-LS** task. The agent encounters an indicator, then must navigate through a series of rooms (see Fig. 3(b)). Each room contains either a silver or blue column, indicating whether the agent must move around it to the left or right to get a reward. At the end, the agent must remember the indicator. There are 16 rooms total, each requiring at least 6 steps to solve. The episode timeout is 200 steps.

**MC Long-Short-Ordered (MC-LSO)** This task tests whether models can learn policies conditioned on distant order-dependencies over two indicators. The two indicators can each be green or red. Only a green followed by red indicates that the goal is to the right at the end. There are 10 rooms with a timeout of 200 steps.

**MC Long-Short-Noise (MC-LSN)** This task starts from **MC-LS** and adds observation noise to test robustness to noise while learning a short and long-term task. For each visual observation we add

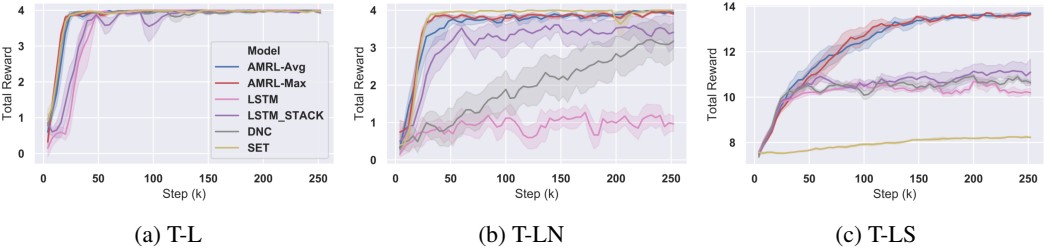

(a) T-L                  (b) T-LN                  (c) T-LS

Figure 4: TMaze Results (5 seeds): AMRL-Max and AMRL-Avg achieve superior performance under observation noise, exploration, and interference short-term tasks. Best viewed in color.

Gaussian noise to each (RGB) channel. An example observation is shown in Figure 3(e). There are 10 rooms with a timeout of 200 steps.

### 4.3 RUNS AND BASELINES

We compare the following approaches: **AMRL-Max.** Our method with the MAX aggregator (Fig. 2d). **AMRL-Avg.** Our method with the AVG aggregator (Fig. 2d). **SET.** Ablation: AMRL-Avg without LSTM or straight-through connection (Fig. 2c). **LSTM.** The currently most common memory model (Hochreiter & Schmidhuber, 1997) (Fig. 2a). **LSTM_STACK.** Stacks two LSTM cells for temporal abstraction (Pascanu et al., 2013; Mirowski et al., 2017) (Fig. 2b). **DNC.** A highly competitive existing baseline with more complex architecture (Graves et al., 2016).

## 5 RESULTS

### 5.1 TMAZE TASKS

Our main results are provided in Figures 4 and 5. We start by analyzing the **T-L** results. In this base task without noise or irrelevant features, we expect all methods to perform well. Indeed, we observe that all approaches are able to solve this task within 50k environment interactions. Surprisingly, the LSTM and stacked LSTM learn significantly slower than alternative methods. We hypothesize that gradient information may be stronger for other methods, and expand on this in Section 6.

We observe a dramatic deterioration of learning speed in the **T-LN** setting, which only differs from the previous task in the additional noise features added to the state observations. LSTM and DNC are most strongly affected by observation noise, followed by the stacked LSTM. In contrast, we confirm that our proposed models are robust to observation noise and maintain near-identical learning speed compared to the **T-L** task, thus validating our modeling choices.

Finally, we turn to **T-LS**, which encapsulates a full RL task with observation features only relevant for the short-term task (i.e. long-term noise induced by short-term features), and noise due to exploration. Our proposed models, AMRL-Avg and AMRL-Max are able to achieve returns near the optimal 13.9, while also learning fastest. All baseline models fail to learn the long-term memory task in this setting, achieving returns up to 10.4.

### 5.2 MINECRAFT TASKS

The **MC-LS** task translates **T-LS** to the visual observation setting. The agent has to solve a series of short term tasks while retaining information about the initial indicator. As before, we see AMRL-Max and AMRL-Avg learn the most rapidly. The DNC model learns significantly more slowly but eventually reaches optimal performance. Our SET ablation does not learn the task, demonstrating that both the order-invariant and order-dependent components are crucial parts of our model.

The **MC-LSO** adds a strong order dependent task component. Our results show that the AMRL-Max model and DNC model perform best here - far better than an LSTM or aggregator alone. We note that this is the only experiment where DNC performs better than AMRL-Max or AMRL-Avg. Here the optimal return is 10.2 and the optimal memory-less policy return is 8.45. We speculate that DNC is able to achieve this performance given the shorter time dependency relative to **MC-LS** and the lower observation noise relative to **MC-LSN**.

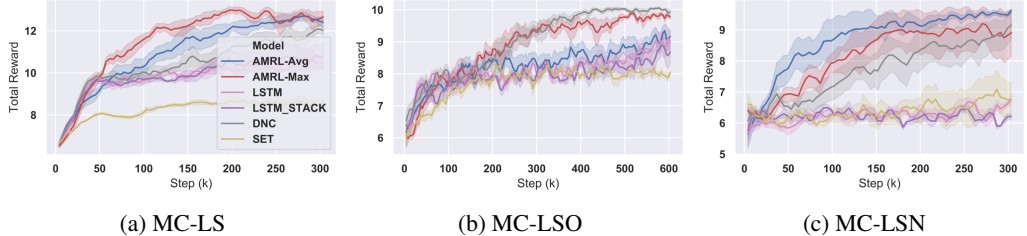

(a) MC-LS  (b) MC-LSO  (c) MC-LSN

Figure 5: Minecraft results (5 seeds): AMRL-Avg and AMRL-Max outperform alternatives in terms of learning speed and final performance.

Finally, **MC-LSN** adds noise to the visual observations. As expected, the LSTM and LSTM_STACK baselines completely fail in this setting. Again, AMRL-Max and AMRL-Avg learn fastest. In this task we see a large advantage for AMRL methods relative to methods with LSTMs alone, suggesting that AMRL has a particular advantage under observation noise. Moreover, we note the strong performance of AMRL-Max, despite the large state-space induced by noise, which affects the SNR bound. DNC is the baseline that learns best, catching up after 300k environment interactions.

## 6 ANALYSIS

Given the strong empirical performance of our proposed method, here we analyze AMRL to understand its characteristics and validate model choices.

### 6.1 PRESERVING GRADIENT INFORMATION

Here we show that the proposed methods, which use different aggregators in conjunction with LSTMs, do not suffer from vanishing gradients (Pascanu et al., 2012; Le et al., 2019), as discussed in Section 3. Our estimates are formed as follows. We set the model input to $1$ when $t = 0$ and to $0$ for timesteps $t > 0$. We plot $avg(d \odot d)$, where $d = 1^T \frac{dg_t}{dx_i}$ over $t$. Samples are taken every ten steps, and we plot the average over three independent model initializations.

Results over time, and for clarity, the final strength of the gradient, are summarized in Figure 6. We observe that the AMRL-Max and AMRL-AVG (and SUM) models have the same large gradient. Our models are followed by DNC[2], which in turn preserves gradient information better than LSTMs. The results obtained here help explain some of the empirical performance we observe in Section 4, especially in that LSTMs are outperformed by DNC, with our models having the greatest performance. However, the gradients are similar when noise is introduced (See Appendix A.4), indicating that this does not fully explain the drop in performance in noisy environments. Moreover, the gradients alone do not explain the superior performance of MAX relative to AVG and SUM. We suggest an additional analysis method in the next subsection.

### 6.2 SIGNAL-TO-NOISE RATIO (SNR)

Following the discussion in Section 3, we now quantify SNR empirically to enable comparison across all proposed and baseline models. We follow the canonical definition to define the SNR of a function over time (Johnson, 2006):

$$SNR_t(f) = SNR(f_t(s_t)), f_t(n_t)) = f_t(s_t)^2 / \mathbb{E}[f_t(n_t)^2] \tag{1}$$

where $t$ denotes time, $f_t$ is a function of time, $s_t$ is a constant signal observed at $t$, and $n_t$ is a random variable representing noise at time $t$. Given this definition, we can derive the SNR analytically for our proposed aggregators (see Appendix A.3 for details).

Analytical results are shown in the last column of Table 1. We see that the AVG and SUM aggregators have the same SNR, and that both decay only linearly. (Empirically, we will see LSTMs induce

---

[2]Gulcehre et al. (2017), analytically show that the gradients of DNC decay more slowly than LSTMs due in part to the external memory writes acting as skip connections or "wormholes".

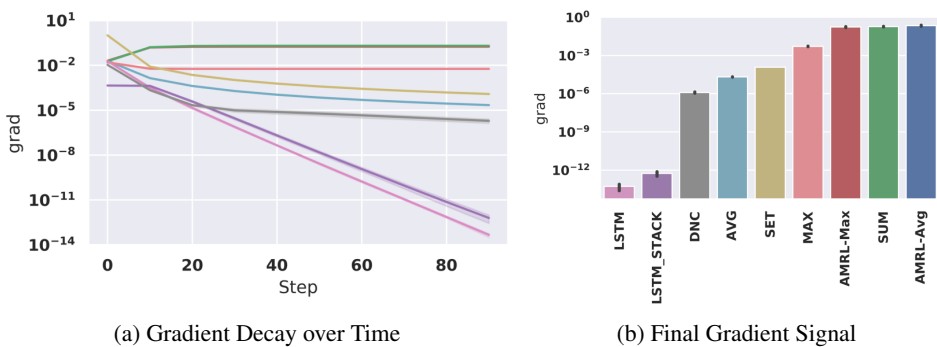

(a) Gradient Decay over Time       (b) Final Gradient Signal

Figure 6: Gradient signal over 0-100 steps. AMRL models and SUM maintain the strongest gradient.

exponential decay.) Moreover, we see that Max has a lower bound that is independent of $t$. Although the bound does depend on the size of the observation space, we observed superior performance even in large state spaces in the experiments (Section 5).

We now turn to an empirical estimate of SNR. In addition to the analytic results presented so far, empirical estimates allow us to assess SNR of our full AMRL models including the LSTM component, and compare to baselines.

Our empirical analysis compares model response under an idealized signal to that under idealized noise using the following procedure. The idealized signal input consists of a single **1** vector (the signal) followed by **0** vectors, and the noisy input sequence is constructed by sampling from $\{\mathbf{0}, \mathbf{1}\}$ uniformly at random after the initial **1**. Using these idealized sequences we compute the SNR as per Eq. 1. We report the average SNR over each neuron in the output. We estimate $E[s^2]$ and $E[n^2]$ over 20 input sequences, for each of 3 model initializations.

The results show that AMRL-Max, Max, and the baseline DNC have the highest SNR. The lowest SNR is observed for LSTM and LSTM_STACK. The decay for both LSTM models is approximately exponential, compared to roughly linear decay observed for all other models. This empirical result matches our derivations in Table 1 for our proposed models. In Figure 7(a), we observe that the SNR for LSTMs strongly depends on the time at which a given signal occurred, while our Max models and DNC are not as susceptible to this issue.

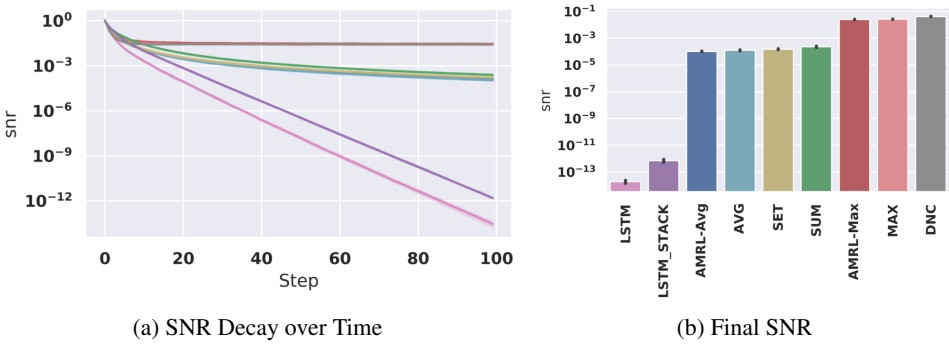

(a) SNR Decay over Time       (b) Final SNR

Figure 7: MAX models and DNC have greatest SNR. LSTM and LSTM_STACK perform worst with exponential decay. SUM and AVG have only linear decay, confirming our analytic finding.

## 6.3 DISCUSSION

The results in the previous section indicate that models that perform well on long-term memory tasks in noisy settings, such as those studied in Section 5, tend to have informative gradients and high SNR over long time horizons. In this section we further examine this relationship.

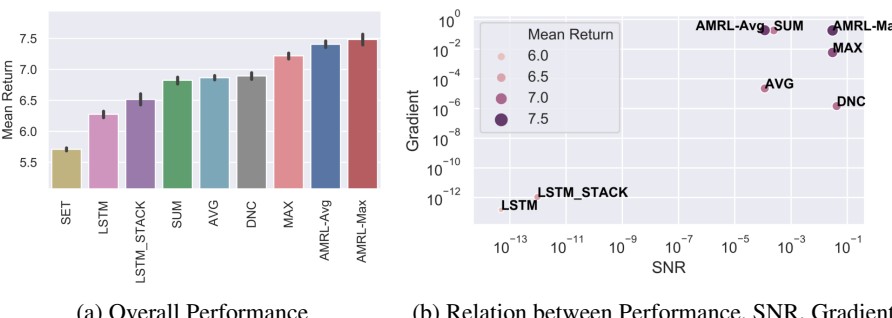

(a) Overall Performance      (b) Relation between Performance, SNR, Gradient

Figure 8: Overall Performance and Performance in relation to SNR and Gradient. Increasing either SNR or Gradient strength tends to increase performance. See text for details on the SUM model.

Figure 8 shows the aggregate performance achieved by each model across the experiments presented in Section 5 and in the appendix A.2. We argue that these tasks capture key aspects of long-term memory tasks in noisy settings. We observe that our proposed *AMRL-Avg* and *AMRL-Max* approaches outperform all other methods. Ablations *Max* and *Avg* are competitive with baselines, but our results demonstrate the value of the ST connection. *AMRL-Max* improves over the LSTM average return by 19% with no additional parameters and outperforms the DNC average return by 9% with far fewer parameters. We have shown that *AMRL* models are not susceptible to the drastic performance decreases in noisy environments that LSTMs and DNCs are susceptible to, and we have shown that this generalizes to an ability to ignore irrelevant features in other tasks.

Figure 8(b) relates overall model performance to the quantities analyzed above, SNR and gradient strength. We find SNR and gradient strength are both integral and complementary aspects needed for a successful model: DNC has a relatively large SNR, but does not match the empirical performance of AMRL – likely due to its decaying gradients. AMRL models achieve high SNR and maintain strong gradients, achieving the highest empirical performance. The reverse holds for LSTM models.

An outlier is the SUM model – we hypothesize that the growing sum creates issues when interpreting memories independent of the time-step at which they occur. The max aggregator may be less susceptible to growing activations given a bounded number of distinct observations, a bounded input activation, or an analogously compact internal representation. That is, the max value may be low and reached quickly. Moreover, the ST connection will still prevent gradient decay in such a case.

Overall, our analytical and empirical analysis in terms of SNR and gradient decay both validates our modeling choices in developing AMRL, and provides a useful tool for understanding learning performance of memory models. By considering both empirical measurements of SNR and gradients we are able to rank models closely in-line with empirical performance. We consider this a particularly valuable insight for future research seeking to improve long-term memory.

## 7 CONCLUSION

We have demonstrated that the performance of previous approaches to memory in RL can severely deteriorate under noise, including observation noise and noise introduced by an agents policy and environment dynamics. We proposed *AMRL*, a novel approach designed specifically to be robust to RL settings, by maintaining strong signal and gradients over time. Our empirical results confirmed that the proposed models outperform existing approaches, often dramatically. Finally, by analyzing gradient strength and signal-to-noise ratio of the considered models, we validated our model choices and showed that both aspects help explain the high empirical performance achieved by our models.

In future research, we believe our models and analysis will form the basis of further understanding, and improving performance of memory models in RL. An aspect that goes beyond the scope of the present paper is the question of how to prevent long-term memory tasks from interfering with shorter-term tasks - an issue highlighted in Appendix A.2.3. Additionally, integration of AMRL into models other than the standard LSTM could be explored. Overall, our work highlights the need and potential for approaches that specifically tackle long-term memory tasks from an RL perspective.

ACKNOWLEDGMENTS

This work was done while Jacob Beck was a research intern at Microsoft Research Cambridge. We would like to acknowledge Adrian O'Grady, Yingzhen Li, Robert Loftin, Quan Vuong, Max Igl, and the Game Intelligence team at Microsoft Research for their discussion, advice, and support.

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

# A APPENDIX

## A.1 ENVIRONMENT DETAILS

In this appendix we provide additional details of all tasks we report on in the main paper.

### A.1.1 TMAZE LONG (T-L)

In these experiments, the agent is initially placed at one end of a corridor. At the other end is a T-junction, where the agent can decide to move left or right. The goal state cannot be observed and is located in one of these two directions. Once the agent chooses to check a given direction, the episode terminates and the agent either receives a success or fail reward, determined by whether the agent picked the direction toward the goal. The position of the goal is randomized between episodes and is indicated only in the very first step by an observation called the *indicator*. **T-L** is our base task where noise is minimized. The agent is automatically moved to the next position in each step to eliminate variation due to an agent's exploratory policy. Thus, there is a single decision to learn - whether to move left or right at the T-junction.

In all TMaze experiments, to avoid confounding factors such as changing the length of BPTT and changing the total number of timesteps per indicator observation, we fix the length of the maze and simply move the indicator. The indicator can be placed at the beginning of the maze or at another location in the middle. The agent receives a reward of 4 for the correct action at the junction and -3 for an incorrect action at the junction. We encode observations as vectors of length 3, with the first dimension taking on the value of 1 if the agent is at the start (0 otherwise), the second dimension taking on the value of 1 or -1 corresponding to the two values of the indicator (when the agent is at the indicator, 0 otherwise), and the final dimension taking on the value of 1 if the agent is at the T-junction (0 otherwise). (For example, [1, -1, 0] encodes an observation for the agent at the start with the goal placed to the right.). Unless otherwise stated, we use a timeout of 150 steps.

### A.1.2 TMAZE - LONG NOISE (T-LN)

Here we append a noise feature to the agent observation to test robustness to observation noise. Noise us sampled uniformly from $\{-1, 1\}$. This experiment is a variant of experiments proposed in Bakker (2001) where continuous valued noise was used. Here we choose discrete noise features as they allow us to build up to the short-long decision task discussed next.

### A.1.3 TMAZE - LONG-SHORT (T-LS)

The short-term task that we add is to "recreate" the noise observation. More precisely, we append a dimension to the action space that allows for two actions: one representing the value 1 and the other representing -1. If this action matches the noise observation, then the agent proceeds to the next step and received a reward of 0.1. Otherwise, the agent stays where it is and the observation is recreated with the noise dimension sampled from $\{-1, 1\}$.

### A.1.4 MINECRAFT MAZE - LONG-SHORT (MC-LS)

The agent starts on an elevated platform, facing a block corresponding to a certain indicator. When on the platform, the agent must step to the right to fall off the platform and into the maze. The agent is now positioned at the southern entrance to a room oriented on the north-south axis. Stepping forward, the agent enters the room. At this point, there are columns to the agent's left and right preventing the agent from moving east or west. The agent has all of the actions (north, east, west) available, but will remain in place if stepping into a column. The agent's best choice is to move forward onto a ledge. On this ledge, the agent faces a column whose block type (diamond or iron) indicates a safe direction (east or west) to fall down. If the agent chooses correctly, it gets a positive reward of 0.1. At this point, the agent must proceed north two steps (which are elevated), fall back to the center, then north to enter the next room. At the very end (purple), the agent must go right if the initial indicators were green (green then red in the multi-step case), and left otherwise. The agent receives a reward of 4 for a correct action at the end and -3 otherwise. In addition, if the agent takes an action that progresses it to the next step, it receives a reward of 0.1 for that correct action.

Unless otherwise stated, we use a timeout of 200. Here 13.7 is optimal, and 10.2 is the best possible for a memory-less policy. There are 16 rooms total, each requiring at least 6 steps each to solve.

### A.1.5 MINECRAFT MAZE - LONG-SHORT ORDERED (**MC-LSO**)

In this version of the Minecraft Maze, there is a second initial platform that the agent will land on after the first and must also step right off of to enter the maze. The options for the colors of the indicators are (green, red), (red, green), (green, green), (red, red). Of these, only the first indicates that the agent is to take a right a the end of the maze. As in the T-Maze Long-Short Ordered environment, the goal here is to see if aggregating the LSTM outputs is capable of providing an advantage over an aggregator or LSTM alone. We use only 10 rooms given the time required to solve this environment. We speculate that this environment is the only one where DNC outperforms our ST models due to the shorter length, which gives our models less of an advantage.

### A.1.6 MINECRAFT MAZE - LONG-SHORT NOISE (**MC-LSN**)

In this version of the Minecraft Maze we start from the **MC-LS** task and add 0-mean Gaussian noise to each channel in our RGB observation, clipping the range of the pixels after the noise to the original range of range [-1,1]. The noise has a standard deviation of 0.05. In addition to adding noise that could affect the learning of our models, this experiment tests learning in a continuous observation space that could be problematic for the MAX aggregator. We use 10 rooms for this experiment with optimal return 10.1, and the optimal memory-less policy return is 6.6.

## A.2 ADDITIONAL EXPERIMENTS

In addition to our primary experiments presented in the main paper, we designed the following experiments to confirm that our proposed models retain the ability to learn *order dependent information*, using the path through the LSTM model. The expected result is that learning speed and final outcome matches that of the baseline methods, and that the SET model cannot learn the order-dependent aspects of the tasks. This is indeed what we confirm.

### A.2.1 LONG ORDER VARIANCE EXPERIMENT (**T-LO**)

This experiment modifies the TMaze such that there are two indicators at opposite ends of the hallway. We place an indicator at position 1 and N-2 (the ends of the corridor that are just adjacent two the start and T-junction respectively). The two indicators take on one of the 4 pairs of values with equal probability: [1, -1], [-1, 1], [1, 1], [-1, -1]. Only the first of these [1, -1] corresponds to the goal being placed to the left at the end. In order to solve this environment optimally, the agent must remember both of the indicators in the correct order.

We expect that a single $h_t$ would need to be used to encode both, due to the order-variance, and that SET cannot perform this task. Given that the indicators on which the order depends span the length of the maze, we do not expect to see any performance differences between methods other than SET. Results are shown in Figure 9 (left) and confirm our hypothesis. SET is not able to learn the task while all other approaches correctly learn order-dependent information and solve the task optimally within at most 50k steps.

### A.2.2 TMAZE LONG-SHORT ORDERED EXPERIMENT (**T-LSO**)

In order to see whether we can learn policies conditioned on distant order-dependencies, along with irrelevant features, we extend the **T-LS** environment with an additional indicator, similar to that in **T-LO**. As above, the two indicators can take on values in the pairs: [1, -1], [-1, 1], [1, 1], [-1, -1], with equal probability. Only the first of these [1, -1] corresponds to the goal being placed to the left at the end. In this experiment, the two indicators were placed at positions 1 and 2, so that their observation does not include the start bit, which could be used to differentiate the two. Unlike in **T-LO**, our indicators appear at the start of the corridor and are adjacent. Here, we expect baseline methods to be less sample efficient than AMRL because gradients decay over long distances.

Our results in Figure 9 confirm our hypothesis. We see only AMRL-Avg and AMRL-Max are able to exceed the return of the best memory-less policy (12.15). We confirm by inspecting the individual

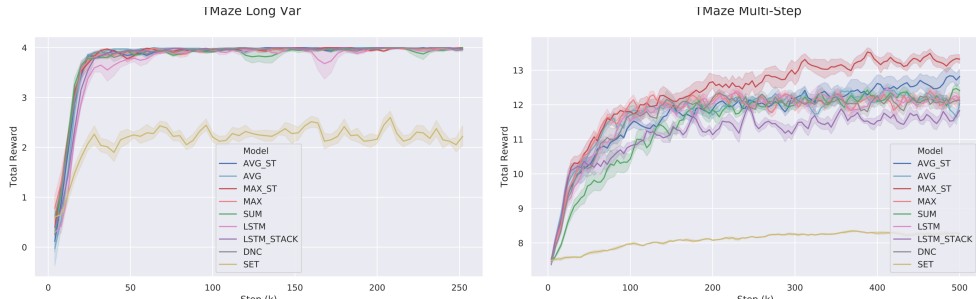

Figure 9: Results: **T-LO** (left) and **T-LSO** (right) experiments. These are included in overall performance but not discussed in the main body. Our results confirm that AMRL models maintain order dependent memories while the SET ablation does not. Note: "ST" here is short for "AMRL".

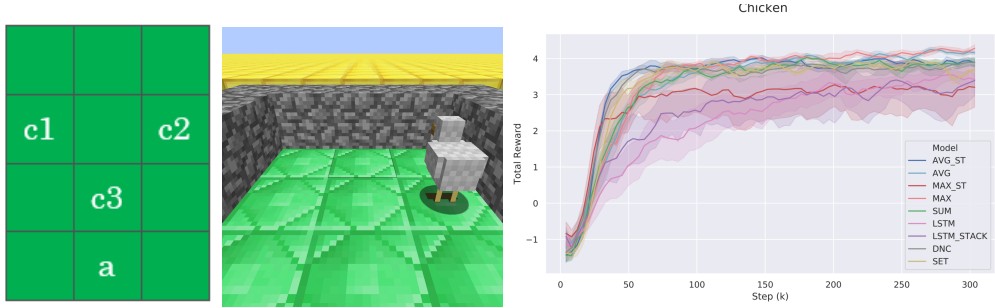

Figure 10: Top down map of Chicken environment (left) and a view from the agent near the start of the episode when the signal is green (center). Results are shown on the right.

learning curves, that both AMRL-Avg and AMRL-Max achieve reward near optimal. This is also the only setting where stacked LSTM performed worse than LSTM.

### A.2.3 MINECRAFT CHICKEN

Our final environment is also constructed in Minecraft and is meant to more closely emulate the Strong Signal setting with a recurring signal, as defined in A.3 below. In this environment, the agent inhabits a 4 by 3 room, starting at the south end in the middle (a on the map), and must collect as many chickens as possible from positions c1, c2, or c3 (see Figure 10, left and center), receiving a reward of 0.1 per chicken. The chicken inhabits one of three locations. The chicken starts forward one block and offset left or right by one block (c1 or c2), placed uniformly at random. The chicken is frozen in place and cannot move. Once collected, a new chicken spawns behind the agent in the middle of the room just in front of the agent's original spawn location (c3). Once that chicken is collected, the next chicken will again spawn forward and offset (c1 or c2), and the cycle continues.

After 48 steps of the repeated signal, the floor changes from red or green to grey. The agent then has another 96 timesteps to collect chickens. At the end, the agent must recall the initial floor color to make a decision (turn left/right). If the agent is correct, it receives a reward of 4 and keeps all the chickens, otherwise it receives a reward of -3 and falls into lava. The agent has 5 actions: forward, backward, left, right, collect. In the final step, when the agent is asked to recall the room color, left corresponds to red, right corresponds to green, and all other actions are incorrect (reward -3).

We see that all models quickly plateau at a return near 4, although 8.8 is optimal. Roll-outs indicate that all models learned to remember room color, but struggled to collect chickens. Training the best performing model, MAX, for 1.5 million steps, we saw still perfect memory in addition to good chicken collection (e.g. roll-out: https://youtu.be/CLHml2Ws8Uw), with most mistakes coming from localization failure in an entirely grey room. Chicken collection can be learned to the same extent, but is learned slightly faster, without the memory-dependant long-term objective.

### A.3 Signal-to-Noise Ratio (SNR) Analytical Derivations

We derive the SNR analytically for our proposed aggregators in two different settings. We term the setting assumed in the main body of the paper the *weak signal setting*. In this setting we assume that the signal $s$ takes on some initial value $s_0$, followed by $'0's$. This simulates the setting where an initial impulse signal must be remembered. Additionally, we assume that each $n_t$ is 0-mean and is sampled from the same distribution, which we call $\rho$.

The assumptions of the weak signal setting are motivated by our POMDP from the introduction in which an agent must remember the passcode to a door. In this setting, the order of states in between the passcode (signal) and the door are irrelevant (noise). Moreover, all of our aggregators are commutative. Thus, instead of the ordered observations in a trajectory, we can consider the set of states the agent encounters. If the episode forms an ergodic process, then as the episode continues, the distribution over encountered observations will approach the stationary distribution $\rho$, which defines a marginal distribution over observations. Thus, it is useful to consider the case where each state is drawn not from $\mathcal{O}(o_t|s_t)$, but rather i.i.d. from $\rho$, for the purpose of analysis.

In addition to the weak signal setting, it is worth considering a recurring signal. We term such a setting the *strong signal setting*. The strong signal setting assumes that the signal recurs and that the signal and noise are not present at the same time. Specifically, we assume that there is one signal observation $o_s$, which can occur at any time; that each observation $o_t$ is drawn from $\rho$; that the signal $s_t$ is $o_s$ if $o_t = o_s$, and 0 otherwise; and that the noise is 0 if $o_t = o_s$, and $o_t$ otherwise. This setting would be relevant, for example, for an agent engaging in random walks at the start of training. This agent may repeatedly pass by a signal observation that must be remembered.

For the weak and strong signal settings, we now analytically derive the SNR of each of our proposed aggregators (summarised previously in Table 1).

**Average Aggregator** Signal averaging is a common way to increase the signal of repeated measurements. In the *weak signal* case, the signal decays, but only linearly. Writing $s$ as a shorthand for $s_0$:

$$SNR(avg(s_t), avg(n_t)) = (\frac{1}{t}s_0 + 0)^2 / \mathbb{E}[(\frac{1}{t}\sum_{i=1}^{i=t} n_i)^2] \text{ (From Eq. 1)}$$

$$= s^2 / \mathbb{E}[(\sum_{i=1}^{i=t} n_i)^2]$$

$$= s^2 / (\mathbb{E}[(\sum_{i=1}^{i=t} n_i)^2] - \mathbb{E}[\sum_{i=1}^{i=t} n_i]^2 + \mathbb{E}[\sum_{i=1}^{i=t} n_i]^2)$$

$$= s^2 / (Var(\sum_{i=1}^{i=t} n_i) + \mathbb{E}[\sum_{i=1}^{i=t} n_i]^2)$$

Given i.i.d.

$$s^2 / (t Var(n) + t^2 \mathbb{E}[n]^2)$$

Assuming 0-mean noise:

$$s^2 / (t Var(n))$$

In the *strong signal setting*, we can actually see linear improvement. In this setting, given that the signal is also a random variable, we use:

$$SNR(s, n) = s^2 / \mathbb{E}[n^2] \tag{2}$$

$$SNR_t(f) = SNR(f_t(s_t)), f_t(n_t)) = f_t(s_t)^2 / \mathbb{E}[f_t(n_t)] \tag{3}$$

where $t$ denotes time, $f_t$ is a function dependent on $f_{t-1}$, $s_t$ is a constant signal given time, and $n_t$ is a random variable representing noise at time $t$. Given this definition we have:

$$SNR_t = \mathbb{E}[avg(s_t^2)] / \mathbb{E}[(\frac{1}{t}\sum_{i=1}^{i=t} n_i)^2]$$

$$= \rho(s)s^2/((1-\rho(s))\frac{1}{t^2}\mathbb{E}[(\sum_{i=1}^{i=t} n \sim \rho)^2])$$

$$= \frac{\rho(s)}{1-\rho(s)}(s^2t^2/\mathbb{E}[(\sum_{i=1}^{i=t} n \sim \rho)^2])$$

From above:

$$\frac{\rho(s)}{1-\rho(s)}(s^2t^2/(tVar(n)))$$

$$= \frac{\rho(s)}{1-\rho(s)}(s^2t/Var(n))$$

Note that if we do not assume that the noise is zero-mean, we can derive the following:

$$\frac{\rho(s)}{1-\rho(s)}(s^2t^2/(tVar(n)+t^2\mathbb{E}[n]^2))$$

This converges to:

$$\frac{\rho(s)}{1-\rho(s)}(s^2/\mathbb{E}[n]^2)$$

**Sum Aggregator** In the weak setting:

$$SNR(sum(s_t), sum(n_t)) = s_0^2/\mathbb{E}[(\sum_{i=1}^{i=t} n_i)^2] = SNR(avg(s_t), avg(n_t))$$

In the strong signal setting:

$$SNR(sum(s_t), sum(n_t)) = \rho(s)(ts)^2/((1-\rho(s))\mathbb{E}[(\sum_{i=1}^{i=t} n \sim \rho)^2]) = SNR(avg(s_t), avg(n_t))$$

Thus in both settings, the SNR is the same as that of the average.

**Max Aggregator** For the max aggregator, we find $SNR(max(s_t), max(n_t))$ to be inadequate. The reason for this is that for $s_t$ as we have previously defined it $max(s_0, ..., s_t) = max(s_0, 0)$ will equal $o_s$ if $o_s > 0$, else 0. However, there is no reason that the signal should vanish if the input is less than 0. Instead, we define $m_t$ to represent the signal left in our max aggregator. We define $m_t$ to be $o_s$ if $\forall_{0<i\leq t}(o_s \geq o_i)$ else 0. That is, if the signal "wins" the max so far, $m_t$ is $o_s = s$, else 0, representing no signal. We define $z_t$ to be $max_{0<i\leq t}(o_t)$ if $\exists_{0<i\leq t}(o_i > o_s)$ else 0. In the continuous setting (weak setting with no repeats of any observation - signal or noise) we have:

$$SNR(m_t, z_t) = \frac{\frac{1}{t+1}s^2 + \frac{t}{t+1}0}{\frac{1}{t+1}0 + \frac{t}{t+1}\mathbb{E}[max_{0<i\leq t}(n_i)^2]} = s^2/\mathbb{E}[max_{0<i\leq t}(n_i)^2] \leq SNR(avg(s_t), avg(n_t))$$

In the discrete setting, we can derive the bound:

$$SNR(m_t, z_t) = \frac{\mathbb{E}[m^2]}{\mathbb{E}[z^2]} \geq \frac{\frac{1}{|\Omega|}s^2}{\frac{|\Omega|-1}{|\Omega|}max(\Omega)^2} = \frac{s^2}{(|\Omega|-1)max(\Omega)^2}$$

which conveniently has no dependence on $t$ and is very reasonable given a small observation space.

## A.4 STRONG SNR AND NOISY GRADIENTS

In this section, we empirically measure the gradient under observation noise and empirically measure the SNR in the strong signal setting. We see that the gradient in the noisy setting is similar to the gradient in the non-noisy setting. We also see that when the signal is attenuated in the strong setting, the SNR of LSTM-based methods drops substantially. This is due to the SNR of the LSTMs being dependent on recency of the signal, and also results in greater variance in the SNR. Our models, on the other hand, have a SNR with low variance and an SNR that is resilient to signal attenuation. Our models have a stronger dependence on the number of prior signals than on the recency of the signal.

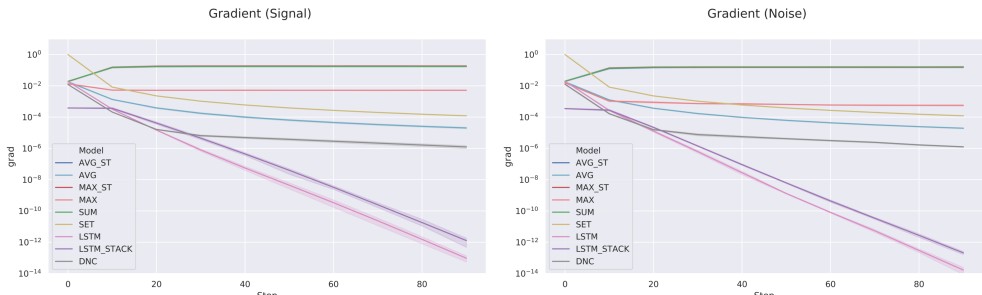

Figure 11: (Left:) Decay of gradient in strong signal setting. (Right:) Decay of gradient when the input is sampled from $U\{1, -1\}$. Results are surprisingly similar to non-noisy setting.

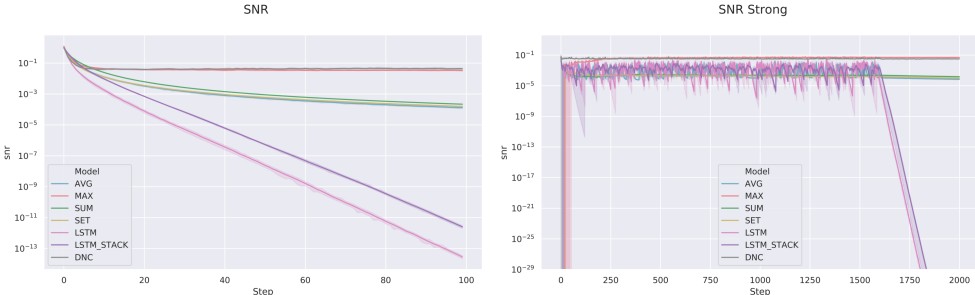

Figure 12: (Left:) SNR of various models in the weak signal setting. (Right:) SNR of models in strong signal setting defined in A.3. LSTM and LSTM_STACK fail when the signal is removed, since they are very dependent on being temporally close to the signal, instead of simply depending on the number of prior signals. Again, MAX and DNC perform the best.

## A.5 HYPER-PARAMETERS

**Learning Rate**  For learning rates, the best out of 5e-3, 5e-4, and 5e-5 are reported in each experiment, over 5 initialization each.

**Recurrent Architecture**  All LSTM sizes were size 256. DNC memory size is 16 (slots), word size 16 (16 floats), 4 read heads, 1 write head, and a 256-LSTM controller.

**Feed-forward Architecture**  We use ReLU activations. Our last feed-forward layer after the memory module is size 256 ($FF_2$). Before the memory module we have two feed forward layers both size 256 ($FF_1$). For models with image input, $FF_1$ consists of 2 convolutions layers: (4,4) kernel with stride 2 and output channels 16; (4,4) kernel with stride 2 and output-channels 32. Then this is flattened and a fully-connected layer of size 256 follows.

**Optimizer**  we use the adam optimizer (Kingma & Ba, 2015), and a PPO agent (Schulman et al., 2017). For training our PPO agent: mini-batch size 200, train batch size 4,000, num_sgd_iter 30, gamma .98.

**Software**  We used Ray version 0.6.2 (Liang et al., 2018). We used python 3.5.2 for all experiments not in the appendix, while python 3.5.5 was used for some earlier experiments in the appendix. Python 3.6.8 was used for some empirical analysis.

## A.6 LSTM DEFINITION

The LSTM maintains its own hidden state from the previous time-step, $h_{t-1}$, and outputs $h_t$. We use the following definition of the LSTM (Hochreiter & Schmidhuber, 1997):

$$\boldsymbol{i}_t = \sigma(\boldsymbol{W}_{i,x}\boldsymbol{x} + \boldsymbol{W}_{i,h}\boldsymbol{h} + \boldsymbol{b}_i)$$
$$\boldsymbol{f}_t = \sigma(\boldsymbol{W}_{f,x}\boldsymbol{x} + \boldsymbol{W}_{f,h}\boldsymbol{h} + \boldsymbol{b}_f)$$
$$\boldsymbol{z}_t = \sigma(\boldsymbol{W}_{o,x}\boldsymbol{x} + \boldsymbol{W}_{o,h}\boldsymbol{h} + \boldsymbol{b}_o)$$
$$\hat{\boldsymbol{c}}_t = tanh(\boldsymbol{W}_{c,x}\boldsymbol{x} + \boldsymbol{W}_{c,h}\boldsymbol{h} + \boldsymbol{b}_c)$$
$$\boldsymbol{c}_t = \boldsymbol{c}_{t-1} \odot \boldsymbol{f}_t + \hat{\boldsymbol{c}}_t \odot \boldsymbol{i}_t$$
$$\boldsymbol{h}_t = tanh(\boldsymbol{c}_t) \odot \boldsymbol{z}_t$$

