# OpenReview forum: "AMRL: Aggregated Memory For Reinforcement Learning"
_ICLR.cc/2020/Conference — Accept (Poster)_

### Official Review · AnonReviewer1 · 2019-10-23
**Official Blind Review #1**

**Rating:** 8

**Review:**

# UPDATE after rebuttal

I have changed my score to 8 to reflect the clarifications in the new draft.

Summary:

This paper presents a family of architectural variants for the recurrent part of an RL controller. The primary claim is that simple components which compute elementwise sum/average/max over the activations seen over time are highly robust to noisy observations (as encountered with many RL environments), as detailed with various empirical and theoretical analyses. By themselves these aggregators are incapable of storing order dependent information, but by combining an LSTM with an aggregator, and pushing half the LSTM activations through the aggregator, and concatenating the the other half of the activations with the aggregator output, the resulting output contains order dependent and order independent content.

The motivation is very clear (many of the most challenging modern video games used as RL environments clearly have noisy observations, and many timesteps for which no new useful information is observed) and the related work is comprehensive. An increase in training speed and stability, apparently without any major caveats, would be of great interest to any practitioner of Deep Reinforcement Learning.

The experiments provided are good, and vary nicely between actual RL runs and theoretical analysis, all of which convinces me that this could well become a standard Deep RL component. I do have a range of questions & requests for clarification (see below) but I believe the experiments as presented, plus some additions before camera ready, will make for a good paper of wide interest.


Desicion: Accept. I would give this a 7 if I was able to. The idea is very simple (indeed I find it slightly hard to believe no one has tried this before, but I don't know of any references myself) and the results, particularly Figure 4, are compelling. It's nice to see a very approachable more mathematical analysis as well, it would be good to see more papers proposing new neural components with this kind of rigour. I look forward to trying this approach myself, post publication.


Discussion:

AMRL-Avg coming out the best in Figure 8 makes a lot of sense to me, as I can see how average provides a stable signal of unordered information. One thing that really doesn't make sense to me is why Max and Sum would also be good - obviously their SNR / Jacobians are quite similar, but fundamentally there is a risk of huge values being built up in activations (especially with Sum), which at least have the potential to cause numerical instability, provide weird gradient magnitudes (which then mess up moment tracking optimizers like Adam, etc). Thre does not seem to be any mention of numerical stability, or whether any considerations ned to be taken for this. Maybe we could hope that 'well behaved' internal neron activations are zero mean, so the average aggregator will never get too big - but is this always the case, at all points in training, in every episode? I appreciate the straight through estimator might ameliorate this, but it is not made entirely clear to me in the text that this is the reason for using it. Addressing this point would increase the strength of the argument.

Given that DNC was determined to be the strongest baseline in a few of the metrics, and that AMRL combines these aggregators with LSTM, an experiment that I'm surprised is missing would be to make AMRL containing a DNC instead of an LSTM. Is there a reason why this wasn't attempted?


It would have been good to see a wider set of RL experiments - the Atari suite is well studied, easily available, and there are many open source implementations to which AMRL could easily be slotted into (eg IMPALA, openai baselines, etc).

The action space for Minecraft is not spelled out clearly - at first I assumed this was the recent project Malmo release, which I assume to have continuous actions (or at least, the ability to move in directions that are not compass points), but while Malmo is mentioned in the paper, the appendix implies that the action space is (north, east, west) etc. I'm aware that there is precedent in the literature (Oh et al 2016) for 'gridworld minecraft', but I think it would improve the paper to at least acknowledge this in the main text, as I feel even most researchers in 2019 would read the text and assume the game to be analogous to VizDoom / DeepMind Lab, when it really isn't. Note that most likely the proposed method would give an even bigger boost with "real" minecraft, as there are even more non-informative observations between the interesting ones, and furthermore I think the environment choice made in this paper is fine, as it still demonstrates the point. A single additional sentence should suffice.

Minor points:

NTM / DNC were not "designed with RL in mind" per se, the original NTM paper had no RL and the DNC paper contained both supervised and RL results. Potentially the statement at the bottom of page 1 was supposed to refer mainly to stacked LSTMs - either way I feel it would be better to slightly soften the statement to "...but also for stacked LSTMs and DNCs (cite), which have been widely applied in RL."

In the RL problem definition, the observation function and the state transition function are stated as (presumably?) probability distributions with the range [0,1], but for continuous state / observation spaces it is entirely possible for the probability density at a point to exceed 1.

Also in the RL section - the notation of $\tau_t \in \Omega^t$ should probably also contain the sequence of $t-1$ actions taken throughout the trajectory - this is made clear in the text, but not in the set notation.

The square bracket "slicing" notation used for slicing and concatenating is neat, but even though I spend all day writing python it still took me a while to realise what this meant, as I haven't seen this used in typeset maths before. Introducing this notation (as well as the pipe for concatenation) at first usage would help avoid tripping up readers unnecessarily.



Note that this pdf caused several office printers I tried it on to choke on later pages, and I get slow scrolling in Chrome on figures such as figure 11 - possibly the graphics are too high resolution (?)


typos: A5, feed forward section: "later" should be "layer" I believe.

**Experience Assessment:**

I have published in this field for several years.

**Review Assessment: Checking Correctness Of Derivations And Theory:**

N/A

**Review Assessment: Checking Correctness Of Experiments:**

I carefully checked the experiments.

**Review Assessment: Thoroughness In Paper Reading:**

I read the paper at least twice and used my best judgement in assessing the paper.

---

> ### Author Response · Authors · 2019-11-09
> **Response to Reviewer 1**
>
> Thank you for your in depth review and constructive feedback. We appreciate you noting the rigour of our research in our combination of experimental and theoretical analysis. We also find it affirming that you state AMRL could become a standard Deep RL component of wide interest.
>
> We address detailed points in turn below:
>
> Discussion:
> 1) Your intuition certainly aligns with our own. In fact, the Sum did not perform as well as would be expected given that is has the same Jacobian and SNR as AMRL-Avg. We discuss this in the discussion section: “An outlier is the SUM model – we hypothesize that the growing sum creates issues when interpreting memories independent of the time step at which they occur”. As for Max, empirically we find Max to outperform the rest. Strong performance despite increasing activations may be due to a bounded number of distinct observations in the discrete setting, bounded input activations, or an analogously compact internal representation. That is, the max value may be low and reached quickly. In either case, we believe that the straight through estimator will ameliorate negative effects on gradient decay. Thank you for noting this. We will add an additional point of clarification to the discussion section.
> 2) We targeted the LSTM in combination with AMRL given that the detrimental effect of noise most needed correcting for LSTMs. Moreover LSTMs are the current standard, whereas DNCs have not quite gained as wide of an acceptance. (Additionally, DNC required more compute and we were already reaching time and compute limitations.) We believe that AMRL insights could be integrated into other models, such as the DNC, and will add this as a potential future direction.
> 3) Almost all settings in the Atari suite are fully-observable, so we did not feel they would offer a useful comparison of memory. Moreover, the present tasks allow us to vary properties of the RL environment and systematically assess properties of the proposed approach and baseline models pertinent to memory.
> 4) You are correct. We are using Malmo, and our version has discrete actions. We will add a sentence.
>
> Minor Points:
> 5) We will change "designed with RL in mind" to “...but also for stacked LSTMs and DNCs (cite), which have been widely applied in RL."
> 6) We will change [0,1] to \R_{\ge 0}.
> 7) In fact, we did not condition our networks on prior actions during training, since it was not relevant for our environments. We will clarify this point.
> 8) Slicing notation: Thanks for pointing out this lack of clarity. We will introduce the notation at first usage in the updated paper.
> 9) Thank you for bringing this to our attention. We have changed each pdf graphic to a png with a reasonable density in hopes of resolving the issue. This drastically reduced the file size. We will include this soon in the updated paper.
> 10) Typo: You are correct and we will make the edit.

---

> > ### Comment · AnonReviewer1 · 2019-11-15
> > **Response to Rebuttal**
> >
> > Thanks for your rebuttal and updated version. With the clarifications added, I am raising my score to 8.

---

### Official Review · AnonReviewer2 · 2019-10-24
**Official Blind Review #2**

**Rating:** 6

**Review:**

UPDATE: The response helped address my questions. I've raised the score to 6.

This paper studies reinforcement learning for settings where the observations contain noise and where observations have long-range dependencies with the past. The proposed approach builds on the LSTM model and adds an aggregated memory cell, which decreases noise by allowing them to cancel at a high level.

Extensive experiments are provided on two sets of tasks, the TMaze and the Minecraft tasks. The experimental results look convincing to me. However, as someone who is outside the area, it is difficult to understand the details of the paper. There are numerous observations and experimental design choices which are not clearly explained I think.
- The current version (~ 10 pages) is significantly over length.
- LSTMs are sensitive to noise: Is there an explanation for this observation? Also, is the claim still true by suitably regularizing the LSTM, etc?
- TMaze Long-Short: What's the intuition for why this setting requires learning over long-term memory tasks?

More detailed comments/questions:
- Intro P1: You start by talking about tasks that require long-term memory. Then you talk about full vs partial observations. What's the connection between these two?
- Intro P4: This observation is interesting -- is there an explanation or intuition for what's happening here?
- Figure 1: Why 68% confidence interval?
- Aggregators: The 1/2 notation in the definition of m_t looks very confusing.
- Definition of a_t, P5: what is FF_2?
- TMaze Long Noise: By adding noise, do you mean that the observations are simply randomly sampled from {-1, +1}?


**Experience Assessment:**

I do not know much about this area.

**Review Assessment: Checking Correctness Of Derivations And Theory:**

N/A

**Review Assessment: Checking Correctness Of Experiments:**

I did not assess the experiments.

**Review Assessment: Thoroughness In Paper Reading:**

I made a quick assessment of this paper.

---

> ### Author Response · Authors · 2019-11-09
> **Response to Reviewer 2**
>
> Thank you for you feedback. We appreciate you noting the strength of the experimental results and appreciate the opportunity to improve the clarity so that our paper can be made more accessible. We address the specific questions you raised below and will incorporate your feedback into the updated version of the paper:
>
> -“LSTMs are sensitive to noise”: Our analysis and discussion sections hypothesize why we see this drop in performance and how to fix it. At a high level, LSTMs are more sensitive to more recent observations. By integrating over fewer (recent) samples, they are more sensitive to slight variations in those samples. Regularization of the inputs may help deal with noise in the observation function, but is unlikely to help with stochasticity in the trajectory. Regularization naively applied to the RNN can be destructive (e.g. dropout. “Recurrent Neural Network Regularization” [ICLR’14])  or require additional learning parameters. In our work we show that, as opposed to changing the objective function, we can address the limitations of RNNs constitutionally and without learning, which give us the sample efficiency critical for RL.
> Moreover, by incorporating all inputs, not only does our approach address prototypical noise in the inputs, but also stochasticity in the trajectory from policy entropy, which is a general challenge we face, given the high degree of exploration almost always necessary at the beginning of training. The benefits we get in terms of noise while not losing - and in fact improving - sensitivity to distant past observations is introduced specifically by order invariant operators. The elegance of solving both problems (stochasticity from multiple sources and gradient decay) at once, with no additional parameters, was the motivation for order invariant aggregators.
>
> -”TMaze Long-Short”: In TMaze experiments, including TMaze Long-Short, the indicator is only observed at the beginning but is required to be remembered until the end of the episode to get the full reward.
>
> -”length”: The current version is 10 pages. It is our understanding that this is the limit and is allowable if necessary. We feel that the extra pages allow for additional figures, discussion, and explanation of our experiments. We highlight the fact that we have many experiments and also many instructive diagrams to aid in visual comprehension.
>
> More detailed comments:
> 1) Intro P1. If an environment is not fully-observable, memory is needed to recall previous observations that may be important. E.g. the color of the indicator in MineCraft experiments is part of the state (and will determine reward at termination), but the only way to know the color is via memory, since this part of the state is not observable from the terminal state. We will clarify this in the updated paper.
> 2) Intro P4. Are you referring to the observations regarding figure 1? If so, our analysis and discussion sections hypothesize why we see this drop in performance and how to fix it. Please also see our response to "LSTMS are sensitive to noise.”
> 3) Figure 1. We use 68% confidence interval to show approximately 1 standard deviation. This makes the plots easy to read while giving a sense of variability of our results.
> 4) Notation: Thanks for pointing out this lack of clarity in the slicing 1/2 notation. We have introduced the notation at first usage in the updated paper.
> 5) FF2: Just as FF1, FF2 are feedforward layers defined in the appendix. We will add a reference similar to the one for FF1 that points to the appendix in the updated paper. Thank you for pointing this out.
> 6) TMaze Long-Noise. Not quite: We append noise in the range {-1, +1} to the observation, which is vector-valued. We will clarify this in the updated paper.

---

> > ### Comment · AnonReviewer2 · 2019-11-15
> > **Response**
> >
> > Thanks for the polite response. I've updated the review score.

---

### Official Review · AnonReviewer3 · 2019-10-25
**Official Blind Review #3**

**Rating:** 6

**Review:**

The paper proposes to add one more memory layer on top of LSTM that offers more direct integration of information over time. The goal is to demonstrate that it is very useful to aggregate previous information in a RNN-based reinforcement learning settings. The set-based aggregation improves the gradient over time and maintain good signal-to-noise ratio. The model is evaluated on Tmazes and Minecraft.

Overall it is an interesting paper in the sense that the introduced aggregation layer is so simple but effective in noisy RL. The explanation using arguments of gradient decay and SNR decay seems to be convincing.

It is interesting to know if stacked LSTMs with vertical skip connection as in ARML work because the upper LSTM seems to have the same design goal of integrating information over time.

**Experience Assessment:**

I have published one or two papers in this area.

**Review Assessment: Checking Correctness Of Derivations And Theory:**

I assessed the sensibility of the derivations and theory.

**Review Assessment: Checking Correctness Of Experiments:**

I assessed the sensibility of the experiments.

**Review Assessment: Thoroughness In Paper Reading:**

I read the paper at least twice and used my best judgement in assessing the paper.

---

> ### Author Response · Authors · 2019-11-09
> **Response to Reviewer 3**
>
> Thank you for your positive feedback on the efficacy of our method and analysis of gradient and SNR decay. We have a similar intuition as to why a stacked LSTM has a greater tolerance to stochasticity. Although LSTMs are generally more sensitive to recent inputs, it is often the purpose of stacking RNNs to change the timescale over which they integrate information, as you mentioned. (e.g. “Training and How to Construct Deep Recurrent Neural Networks.” [ICLR’14].) Our analysis highlights how noise will decrease as observations cancel over longer timescales. We emphasise that although stacking confers an advantage (as we empirically demonstrate) the SNR and gradient still decay rapidly over time for memory models that favor recent observations - unlike our commutative aggregators, which are order-invariant. Thus the improvement from AMRL.

---

### Decision · Program_Chairs · 2019-12-19

**Decision:**

Accept (Poster)

**Comment:**

This paper introduces a way to augment memory in recurrent neural networks with order-independent aggregators. In noisy environments this results in an increase in training speed and stability. The reviewers considered this to be a strong paper with potential for impact, and were satisfied with the author response to their questions and concerns.